# Effect of Maize Conservation Crops Associated with Two Vegetal Covers on the Edaphic Macrofauna in a Well-Drained Savanna of Venezuela

**Jimmy A. Morales-Márquez** [1,*] **, Raimundo Jiménez-Ballesta** [2] **, Rosa M. Hernández-Hernández** [3] **,**
**Gloria K. Sánchez** [4] **, Zenaida Lozano** [5] **and Ignacio Castro** [3]

1 Ecological World for Life, 28003 Madrid, Spain
2 Department of Geology and Geochemistry, Autonoma University of Madrid, 28049 Madrid, Spain; raimundo.jimenez@uam.es
3 IDECYT, Universidad Simón Rodríguez, Caracas 1050, Venezuela; rhernandez@reacciun.ve (R.M.H.-H.); icastro@uners.edu.ve (I.C.)
4 Centro de Estudios Ambientales, Universidad Bolivariana de Venezuela, Caracas 1041, Venezuela; ksanchez@ubv.edu.ve
5 Instituto de Edafología, Universidad Central de Venezuela, Caracas 1040, Venezuela; lozanoz@agr.ucv.ve
* Correspondence: j.morales@ewltech.eu; Tel.: +34-652-644-918

**Abstract:** Conventional agricultural in the Venezuelan Llanos has generated gradual soil degradation. Therefore, conservationist agriculture has been proposed. According to several works, this type of management favors soil macrofauna. To test this hypothesis, the response of soil macrofauna to the establishment of conservationist maize crops, associated with *Brachiaria dictyoneura* (Bd) and *Centrosema macrocarpum* (Cm), was evaluated. The samples of soil and soil macrofauna were taken per vegetation cover at different climatic season over 2 years and 10 months. For this period and under the conditions studied, the results partially refute the hypothesis; on the one hand, they showed that the soil macrofauna of a natural savanna (NS) is the most diverse and equitable ($N_1 = 4.5 \pm 2.8$), followed by the cultivation of maize associated with Cm ($N_1 = 3.2 \pm 1.9$) and the least diverse with Bd ($N_1 = 2.6 \pm 2.1$). Additionally, some taxonomic groups apparently did not tolerate soil intervention, while Termitidae was favored. On the other hand, the temporal variation of the soil macrofauna did not differ between vegetation covers (F: 1.18; $p = 0.37$). This variation could be due to the decrease in TP ($r = -0.55$) and increased BD ($r = 0.56$).

**Keywords:** soil macrofauna; community analysis; agroecology; maize; improved pastures; tropical savannas

## 1. Introduction

In the first decade of the 21st century, Venezuela experienced a demographic increase that raised food demand and pressure on its ecosystems. The savanna of the Venezuelan central llanos recorded, for maize and rice crops, an increase of more than 68% (from 620,869 to 1,043,291 ha) of cultivated hectares [1]. However, the soils of these savannas are well drained and acidic, with high exchangeable aluminum saturation and low organic matter content and nutrients (especially nitrogen and phosphorus) [2], which make them unfavorable for crops [3,4]. Therefore, these agricultural activities have been sustained with an intensive use of fertilizers, pesticides and mechanization under a conventional tillage system [5,6]. These activities, combined with climatic factors (marked drought and strong and erratic rains) and annual pasture fires, have contributed to gradual soil degradation [5,7,8].

This environmental degradation, generated by socio-economic growth, poses one of the most important challenges of today's society: to meet its nutritional needs while preserving the production capacities of agroecosystems for future generations. For this

reason, it is necessary to favor the implement conservation practices, oriented mainly toward minimum tillage and adequate management of crop residues in the soil [7]. This not only maintains the soil's abiotic properties (organic matter, cation exchange capacity, phosphorus, nitrogen, and others) and their biological characteristics (microbial biomass and fauna) before cultivation, but it can improve them or even recover them from a degradation by a conventional crop [2,9,10].

However, to achieve sustainability, it is necessary to know all the dimensions that converge in conservation agriculture. One of them is soil biota, especially edaphic macrofauna (invertebrates larger than 2 mm [11]). This is due to their role in the functioning of ecosystems, specifically in pedogenesis, soil structure, soil nutrient cycle and its fertility [12–15]. Knowing this, it is to be assumed that edaphic macrofauna constitute an important component of soil and play a significant role in soil sustainable productivity. At the same time, the close relationship between soil macrofauna and the physical–chemical and biological properties of the soil makes them sensitive to spatio-temporal variations of these edaphic properties [16,17], particularly to porosity, bulk density [18] and nutrient and/or organic matter content [19,20]. This relationship has led to an increasing interest in knowing the biology and ecology of the soil macrofauna and its function [21,22]. This information has been used in several areas, for example in environmental impact studies, as biological indicators [23–25], restoration of degraded soils [19,26,27] and biological pest controllers [27,28].

Studies and proposals about conservation agriculture have given special importance to edaphic macrofauna. This is because a minimal disturbance in the soil can affect it and, reciprocally, soil macrofauna can affect soil properties [10,18,29]. Thus, better soil conditions would imply greater availability of food, ecological niches and stability over time [16,30,31]. Consequently, there would be greater diversity and greater biological activity in the soil [32,33], thus affecting its physical, chemical and biological properties and thus its fertility [29,34–36].

However, despite the interest in knowledge of edaphic macrofauna [37], several of their biological and ecological aspects are not yet clearly understood [38,39]. This situation limits the generation of sustainable agroecological strategies, which take advantage of the benefits of these organisms and minimize the risks they may present to disturbances in their habitats, for example, certain potentially pest taxonomic groups [40,41]. Therefore, keeping in view the close relationship between the soil macrofauna and the edaphic properties, the aim of this work was to know the response of soil macrofauna to the establishment of maize conservation crops (*Zea mays*, Linnaeus) associated with *Brachiaria dictyoneura* (Figari and De Not.) and *Centrosema macrocarpum* (Benth) in a well-drained savanna of the Venezuelan llanos. It was hypothesized that this agroecological management would favor the community of edaphic macrofauna, manifesting in its ecological attributes: density, richness and diversity of families. To test this hypothesis, the following specific objectives were established: to determine the effects of agroecological management on the taxonomic structure and ecological attributes of the macrofauna of the soil and evaluating possible differences between the communities present in natural savanna soils and cultivated plots by depth and in the temporal gradient; on the other hand, to evaluate the influence of the physical–chemical properties of the soil in these differences.

## 2. Materials and Methods

### 2.1. Study Site Description

The experiment was conducted in the experimental station La Iguana, geographically located at 8.3916° and 8.475° N and 65.4675° and 65.3805° W in the southeast savannas of Guárico state, Venezuela, between 80–120 m above sea level [42]. The area was selected due to its importance at points of agricultural production and where the area's inhabitants depend on the production of maize. The climate is marked by a well-differentiated dry period from November to May and a rainy period between June and October. It has an average annual precipitation of 1369 mm and monthly average temperatures range

between 26 °C and 30 °C (isohyperthermy) [43]. The station, which occupies approximately 3000 hectares, has smooth undulation with 0–2% slope and a soil mosaic with a fertility between low and medium levels; it is slightly acidic, which defines the variety of plant units dominated by the grass *Trachypogon vestitus* (Andersson) [44]. The soil where the research was conducted was classified as Ultisols: Typic Plinthustults [45] with a coarse loam texture and isohyperthermic. A previous study in this area showed that the sand content in these soils is higher than 80%, with a strong acidic reaction, low salinity, low content of organic matter and low nutrient-holding capacity, specially of P and Ca (Table 1) [46]. These areas have been normally used as extensive holdings of low grazing productivity [47].

**Table 1.** Initial physicochemical characteristics of soil studied (*n* = 108 per deep) at the La Iguana experimental station, Guárico state, Venezuela.

| Parameter | Depth (cm) | | |
|---|---|---|---|
| | 0–5 | 5–15 | 15–30 |
| Clay [<2 μm] [†] (%) | 2.50 ± 0.38 [‡] | 2.94 ± 0.09 | 10.00 ± 1.20 |
| Silt [2–5 μm] (%) | 12.51 ± 1.15 | 11.00 ± 0.98 | 12.00 ± 1.02 |
| Very fine sand [50–100 μm] (%) | 6.99 ± 0.32 | 3.17 ± 0.89 | 10.00 ± 1.92 |
| Fine sand [100–250 μm] (%) | 48.91 ± 5.15 | 29.05 ± 2.13 | 36.00 ± 3.16 |
| Medium sand [250–500 μm] (%) | 25.29 ± 1.12 | 44.59 ± 3.32 | 22.93 ± 2.16 |
| Coarse sand [500–1000 μm] (%) | 3.29 ± 0.78 | 7.98 ± 1.12 | 8.20 ± 0.32 |
| Very coarse sand [1000–2000 μm] (%) | 0.51 ± 0.11 | 1.27 ± 0.08 | 0.78 ± 0.10 |
| Textural class | Loamy Sand | Loamy Sand | Sandy Loam |
| Reaction of the soil (pH in $H_2O$) | 5.01 ± 0.18 | 4.81 ± 0.25 | 4.75 ± 0.24 |
| Total acidity ($cmol^+ \cdot kg^{-1}$) | 0.46 ± 0.21 | 0.83 ± 0.58 | 1.18 ± 0.73 |
| Interchangeable aluminum ($cmol^+ \cdot kg^{-1}$) | 0.16 ± 0.09 | 0.34 ± 0.23 | 0.50 ± 0.31 |
| Interchangeable hydrogen ($cmol^+ \cdot kg^{-1}$) | 0.30 ± 0.18 | 0.49 ± 0.42 | 0.67 ± 0.53 |
| Electrical conductivity ($\mu S \cdot cm^{-1}$) | 27.53 ± 1.32 | 23.62 ± 3.26 | 22.77 ± 3.62 |
| CEC ($cmol^+ \cdot kg^{-1}$) | 2.21 ± 0.54 | 1.94 ± 0.66 | 1.90 ± 0.82 |
| Organic matter (%) | 1.33 ± 0.30 | 1.23 ± 0.26 | 1.04 ± 0.27 |
| Total nitrogen (%) | 0.039 ± 0.007 | 0.032 ± 007 | 0.028 ± 0.007 |
| Inorganic nitrogen ($mg \cdot kg^{-1}$) | 21.34 ± 11.18 | 17.23 ± 8.45 | 15.73 ± 9.54 |
| Phosphorus ($mg \cdot kg^{-1}$) | 11.30 ± 0.30 | 10.01 ± 3.62 | 8.67 ± 3.26 |
| Potassium ($mg \cdot kg^{-1}$) | 29.94 ± 14.10 | 19.51 ± 7.31 | 11.53 ± 4.40 |
| Calcium ($mg \cdot kg^{-1}$) | 89.64 ± 25.21 | 63.15 ± 27.87 | 38.11 ± 14.76 |
| Magnesium ($mg \cdot kg^{-1}$) | 51.35 ± 16.94 | 47.19 ± 12.07 | 39.11 ± 14.66 |
| Sodium ($mg \cdot kg^{-1}$) | 2.09 ± 1.33 | 2.40 ± 1.65 | 3.11 ± 1.86 |
| Iron ($mg \cdot kg^{-1}$) | 44.61 ± 22.07 | 53.36 ± 25.03 | 54.38 ± 24.96 |
| Copper ($mg \cdot kg^{-1}$) | 0.71 ± 0.44 | 0.90 ± 0.60 | 1.07 ± 0.57 |
| Manganese ($mg \cdot kg^{-1}$) | 7.76 ± 3.60 | 3.52 ± 2.64 | 2.63 ± 1.73 |
| Zinc ($mg \cdot kg^{-1}$) | 1.07 ± 0.57 | 0.86 ± 0.37 | 0.79 ± 0.39 |

[†] Size of the aggregates. [‡] Standard deviation. Adapted from Hernández et al., 2011 [48].

### 2.2. Design of the Study

Once the site inside the experimental station was selected, an analysis of the space variability of the soil was made in order to define the size of the study plot, its orientation and the number of samples to make [49]. It was concluded that the plot for the natural savanna (NS) had 2 ha (100 m × 200 m) and that both plots with *Brachiaria dictyoneura* (Bd) and *Centrosema macrocarpum* (Cm) had 2.6 ha (350 m × 75 m) [46], each separated by 18 m. Within the plots with Bd and Cm, subplots of 350 m × 18 m were taken for the maize crop without fertilization. In order to have three replicates and generate a stratified random sampling, the plots were in turn subdivided into 3 experimental units of 60 m × 15 m, arranged randomly within the plot, while in the natural savanna was divided equally [48]. Sampling was carried out for climatic seasons over 2 years and 10.5 consecutive months (from 2005 to 2008), synchronizing with climatic periods and with some phases of the maize crop cycle (Table 2). Additionally, for each of the sampling time, samples of mix soil were also taken at different depths in the soil profile, dividing it in depth horizons,

with the average wide criteria of the A horizon (0–15 cm) and the E horizon (15–30 cm). From the A horizon, a layer was taken from 0 to 5 cm deep to evaluate the changes in the edaphic properties, product of the litter and root system of the grass. Thereby, the soil depths studied were 0–5, 5–15 and 15–30 cm [4]. A mixed soil sample consisted of 4 soil sampling points per experimental unit. Therefore, per vegetation cover, there were 3 samples of mixed soil (12 sampling points) per each depth.

**Table 2.** Distribution of sampling times for the study of soil macrofauna and soil of a natural savanna and maize crop associated with *Brachiaria dictyoneura* and *Centrosema macrocarpum* in La Iguana station, Guárico state, Venezuela.

| Descriptor | Sampling Times | | | | | | |
|---|---|---|---|---|---|---|---|
| Days after initiation | 0 | 76 | 188 | 363 | 461 | 678 | 1035 |
| Climate season | Start of rainy season | Rainy season | Dry season | Start of rainy season | Rainy season | Dry season | Dry season |
| Chronological order | T1 | T2 | T3 | T4 | T5 | T6 | T7 |

### 2.3. Soil Preparation for Crops

The establishment of the coverage introduced was conducted by conventionally preparing the soil with two crossed traverse passes, followed by phosphorus rock (Fosfopoder ®: 33% $P_2O_5$) at a rate of 300 kg·ha$^{-1}$, covering it with a trailing pass. Seeding of Bd seeds was performed at the rate of 4 kg·ha$^{-1}$ and Cm at 3 kg·ha$^{-1}$, burying them with a trailing pass. The NS plot was given the proper management of this region, with annual burning (at the beginning of the rainy season) [4,46]. All plots, including Bd and Cm, were grazed with cattle (3 animal units per hectare) twice a year: once at the beginning of the rains and again at the end of the rains [46]. For the cultivation of maize, seeds of the hybrid variety IMECA 3005 were used. The soil of the plots with Bd and Cm was prepared by cutting the coverings flush with the surface of the soil with a rotary and leaving their residues on the ground. Direct seeding was performed with a three-row SEMEATO® machine, at a density of approximately 60,000 plants·ha$^{-1}$ [48].

### 2.4. Soil Macrofauna Sampling

Inside each experimental unit, 2 sampling points were selected randomly. According to the Tropical Soil Biology and Fertility Program [50], every sample consisted of a monolith of 25 × 25 × 30 cm divided in the before-mentioned strata (Figure 1) from where the macrofauna were with direct manual sampling techniques [51]. The macrofauna was taken and preserved in ethylic alcohol (70%) vials. This macrofauna was transferred to the Animal Biology Laboratory, Science Faculty—Universidad de Los Andes (Venezuela), where it was separated and classified taxonomically down to the family level. This identification was made using standardized taxonomic keys in the literature (e.g., Triplehorn et al. 2005) [52]. The collected invertebrates belonging to the taxonomic group of winged insects corresponded to organisms in their larvae stages.

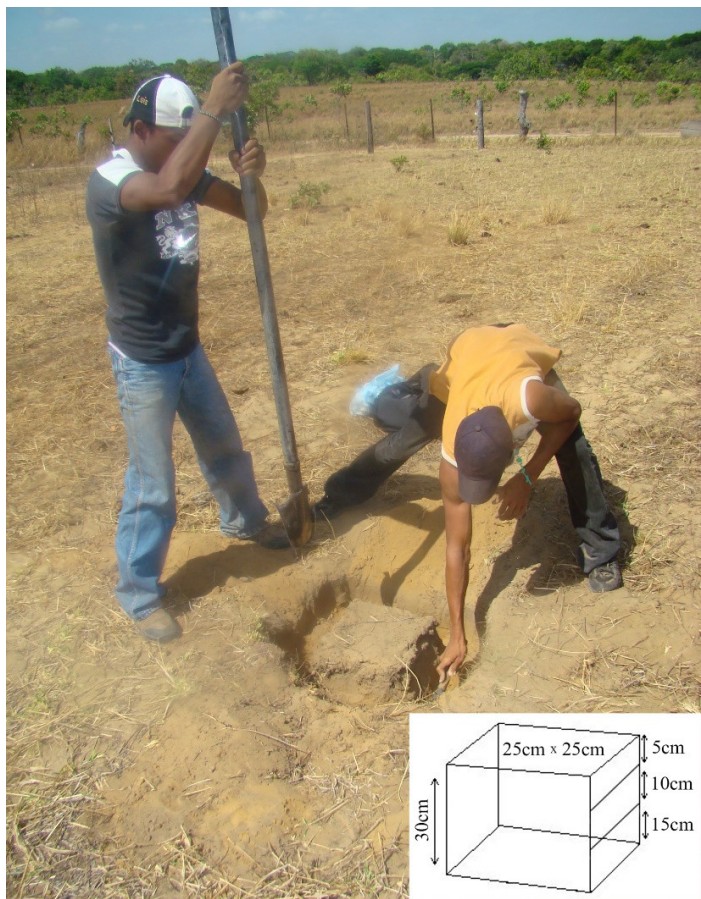

**Figure 1.** Extraction of the soil monolith and representation of the sampling unit.

*2.5. Soil Sampling*

Inside each subplot, four mix soil samples were taken in zig-zag in the before-mentioned depths (Figure 2). For the physical analysis (procedures described in [4,53]),non-altered mix soil samples in 5 × 5 cm cylinders were taken.

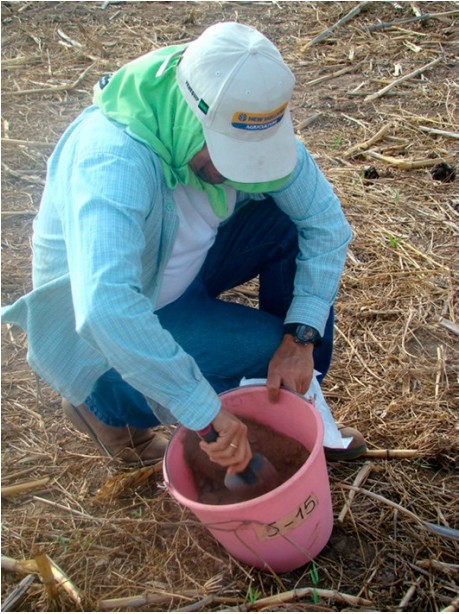

**Figure 2.** Mix soil sample preparation.

*2.6. Laboratory Analysis*

Soil moisture percentage (M) was measured with the gravimetric method. The bulk density (BD) was measured using the method of the cylinder. The pore size was determined with desktops of saturation voltage (TP: total porosity) in a matric potential of $-10$ kPa ($P_{macro}$: radius of the pores > 15 µm.) The retention's porosity ($P_{micro}$) was measured as a result of the difference TP—$P_{macro}$ [54]. The saturated hydraulic conductivity was measured in a constant charge disc permeameter [55]. For the chemical analysis, according to Lozano et al. (2010) [4], the pH was analyzed, measuring the total acidic-AT and electrical conductivity (EC) in a soil–water reaction of 1:1. The organic matter (OM) was studied with the Heanes method [56], oxidizing the organic carbon of the soil sample with potassium dichromate ($K_2Cr_2O_7$) in acidic medium ($H_2SO_4$) and later analyzed through a spectrophotometer. The available phosphorus (P) in the soil was extracted with the Olsen solution [57] and detected with the colorimetric method.

*2.7. Statistical Processes and Calculations*

The families' diversity was analyzed with the $N_0$ (families richness) and $N_1$ from Hill series [58]. To compare the average of the density, richness and diversity values, and the interaction among these factors, the analysis of variance was used. When the result was significant, the differences among the average pairs were determined using the after-test "least significant difference" (LSD) Fischer test [59]. In order to evaluate possible relationships between soil's physical and chemical variables and edaphic macrofauna, we performed, on the one hand, Pearson's linear correlations with STATISTIC software, version 6 [60]; and, on the other hand, a multivariate analysis, with the software CANOCO for Windows, version 4.5 [61]. For this, it was decided to use the linear method "redundancy analysis" (RDA), since the maximum "gradient length" was less than 3 [62]. For all RDAs, the density of families was standardized with their error of variance. The Spearman minimum correlation value [63] was used, plotting a circle of significance ($r_s = 0.38$, $n = 21$ and $p < 0.05$). In all the studies, only the analysis with a $p$ value less than 0.05 were significant. The assumption of normal distribution of the data was checked with the Shapiro–Wilk test [63]. To adjust the normal distribution of the data, the density and families' richness data of the soil macrofauna were transformed with the square root of the value ($x^{1/2}$) and the families' diversity with the square root of the value plus one sixth ($x^{1/2}+ 1/6$). In addition, the $P_{macro}$ was performed with the square root of the value plus four: $(x + 4)^{1/2}$; the Ksat with the square root of the value plus one sixteenth: $(x + 1/16)^{1/2}$; the OM with logarithm of the value plus one ($Log_{10}(x + 1)$) and the EC with the square root of the logarithm of the value: $(\log(x))^{1/2}$.

## 3. Results and Discussion

*3.1. Effect of Agroecological Management on the Structure of the Edaphic Macroinvertebrate Community*

From a global perspective, the abundance and diversity of soil macrofauna found in this study was much lower than reported for humid tropical soils and temperate ecosystems [64–66]. This could be explained due to edaphic (poor soils with low moisture retention) and climatic (high temperatures) factors, which could be unfavorable for a large number of soil macrofauna species [18].

The results reveal differences in the taxonomic structure of soil macrofauna communities associated to plant cover, in terms of quantity and composition of orders and families (Table 3). The natural savanna registered 10 orders with 31 families, while the maize crop associated with Bd coverage registered 11 orders with 30 families, and for Cm coverage, 9 orders and 25 families. In Bd, the orders Orthoptera and Psocoptera appeared, which were absent in NS. While Coccinelidae, Dynastidae (Coleoptera), Cecidyomidae (Diptera) and Pieridae (Lepidoptera) were only present in NS.

**Table 3.** Density (ind·m$^{-2}$) of orders and families, average density, richness, diversity of families of the soil macrofauna of the natural savanna (NS) and maize crop associate with *Brachiaria dictyoneura* (Bd) and *Centrosema macrocarpum* (Cm) in the La Iguana station, Guárico state, Venezuela.

| Order | Vegetation Cover | | | Family | Vegetation Cover | | |
|---|---|---|---|---|---|---|---|
| | NS | Bd | Cm | | NS | Bd | Cm |
| | Ind·m$^{-2}$ | | | | Ind·m$^{-2}$ | | |
| Blattodea (Isoptera) | 30.5 ± 46.4 [†] | 162.3 ± 158.3 | 92.6 ± 89.3 | Termitidae | 30.5 ± 46.4 | 161.5 ± 159.2 | 92.6 ± 89.3 |
| Coleoptera | 77.5 ± 52.8 | 48.4 ± 47.7 | 45.1 ± 38.1 | Carabidae | 18.5 ± 18.3 | 20.2 ± 18.1 | 10.7 ± 10.2 |
| | | | | Staphylinidae | 12.2 ± 16.3 | 8.6 ± 8.1 | 7.4 ± 7.9 |
| | | | | Scarabaeidae | 13.7 ± 17.0 | 5.3 ± 8.1 | 3.8 ± 5.2 |
| | | | | Aphodiidae | 5.5 ± 8.9 | 4.6 ± 4.4 | 11.8 ± 9.9 |
| | | | | Tenebrionidae | 5.1 ± 6.9 | 2.3 ± 4.1 | 3.4 ± 5.2 |
| | | | | Rutelidae | 4.0 ± 6.5 | 2.7 ± 4.3 | 1.9 ± 3.4 |
| | | | | Elateridae | 3.6 ± 5.5 | 1.9 ± 3.3 | 1.5 ± 2.8 |
| | | | | Chrysomelidae | 3.4 ± 5.6 | 0.2 ± 0.4 | 1.7 ± 3.1 |
| | | | | Hydroscaphidae | 2.1 ± 3.8 | 1.7 ± 2.9 | 0.8 ± 1.5 |
| | | | | Lampyridae | 2.9 ± 4.6 | 0.2 ± 0.4 | 0 |
| | | | | Cerambycidae | 1.7 ± 3.1 | 0 | 1.0 ± 1.7 |
| | | | | Coccinelidae | 2.3 ± 4.1 | 0 | 0 |
| | | | | Geotrupidae | 1.0 ± 1.8 | 0 | 1.1 ± 2.1 |
| | | | | Dynastidae | 1.5 ± 2.8 | 0 | 0 |
| | | | | Curculionidae | 0 | 0.8 ± 1.5 | 0 |
| Hymenoptera | 66.5 ± 64.1 | 25.5 ± 24.9 | 45.1 ± 41.9 | Formicidae | 57.7 ± 71.3 | 23.8 ± 22.5 | 28.8 ± 27.3 |
| Haplotaxida | | | | Tenthredinidae | 7.2 ± 11.0 | 1.0 ± 1.7 | 11.8 ± 10.9 |
| | | | | Larva (NI) [‡] | 1.5 ± 2.8 | 0.8 ± 1.5 | 4.6 ± 4.8 |
| | | | | Glossoscolecidae | 19.8 ± 23.0 | 19.2 ± 12.9 | 17.5 ± 16.5 |
| Diptera | 9.9 ± 12.3 | 6.9 ± 10.4 | 7.8 ± 13.0 | Muscidae | 6.9 ± 10.4 | 3.0 ± 5.5 | 3.2 ± 3.9 |
| | | | | Larva (NI) | 1.5 ± 2.8 | 3.0 ± 5.5 | 2.3 ± 4.1 |
| | | | | Sciaridae | 0.8 ± 1.5 | 0.8 ± 1.5 | 2.3 ± 1.4 |
| | | | | Cecidyomidae | 0.8 ± 1.5 | 0 | 0 |
| Hemiptera | 5.1 ± 7.8 | 4.0 ± 6.5 | 4.0 ± 6.9 | Miridae | 3.6 ± 5.9 | 1.0 ± 1.7 | 4.0 ± 5.9 |
| | | | | Cercopidae | 1.1 ± 2.1 | 3.2 ± 5.6 | 0.8 ± 1.5 |
| | | | | Lygaeidae | 1.5 ± 2.8 | 3.0 ± 5.5 | 0 |
| Araneae | | | | Pentatomidae | 0 | 0.8 ± 1.5 | 0 |
| | 6.5 ± 9.3 | 2.3 ± 3.9 | 2.5 ± 4.2 | Paratropididae | 4.2 ± 6.4 | 1.5 ± 2.8 | 2.5 ± 4.2 |
| | | | | Dipluridae | 2.3 ± 4.1 | 0.8 ± 1.5 | 0 |
| Chilopoda | 1.5 ± 2.8 | 0.8 ± 1.5 | 2.3 ± 3.9 | Cryptopidae | 1.5 ± 2.8 | 0.8 ± 1.5 | 1.5 ± 2.8 |
| | | | | Scolopendridae | 0 | 0 | 0.8 ± 1.5 |
| Solifugae | | | | Ammotrechidae | 2.1 ± 3.8 | 1.7 ± 1.9 | 0 |
| Blattodea | | | | Blattelidae | 0 | 0.8 ± 1.5 | 1.5 ± 2.8 |
| Lepidoptera | | | | Pieridae | 0.8 ± 1.5 | 0 | 0 |
| Orthoptera | | | | Gryllotalpidae | 0 | 0.8 ± 1.5 | 0 |
| Psocoptera | | | | Psocidae | 0 | 0.8 ± 1.5 | 0 |
| | | | | Density | 221.3 ± 166.6 | 276.6 ± 266.5 | 205.5 ± 155.4 |
| | | | | Families | | | |
| | | | | Richness | 6.0 ± 3.9 | 4.2 ± 3.6 | 4.6 ± 3.0 |
| | | | | Diversity | 4.5 ± 2.8 | 2.6 ± 2.1 | 3.2 ± 1.9 |

[†] Standard deviation. [‡] Larva Unidentified.

However, despite these differences between plant cover (NS and agricultural management), the one-way analysis of variance shows that average densities of the main families did not present significant differences between these coverages ($p > 0.05$). It was the same for the average of the ecological attributes (density, richness and diversity of families) of the entire soil macrofauna community. It is important to note that this analysis considers the average of the whole community and does not consider which families are or are not present in coverage or their relative densities.

In this sense, the community of soil macrofauna in each vegetation cover was analyzed in terms of its taxonomic structure (Figure 3). This allowed us to verify changes in the relative abundance of the edaphic macrofauna orders, regarding the vegetal cover. It is noteworthy that Isoptera goes from being the third most abundant, with 14% in NS being the main in Bd and Cm, with 59% and 42%, respectively. Meanwhile, Coleoptera was the most dominant in NS, followed by Hymenoptera, with 35% and 30%, respectively,

becoming second and third in cultivated soils. Furthermore, the order Haplotaxida (in this study, endogeic earthworms) was apparently not as affected by agricultural management compared with the three most dominant orders.

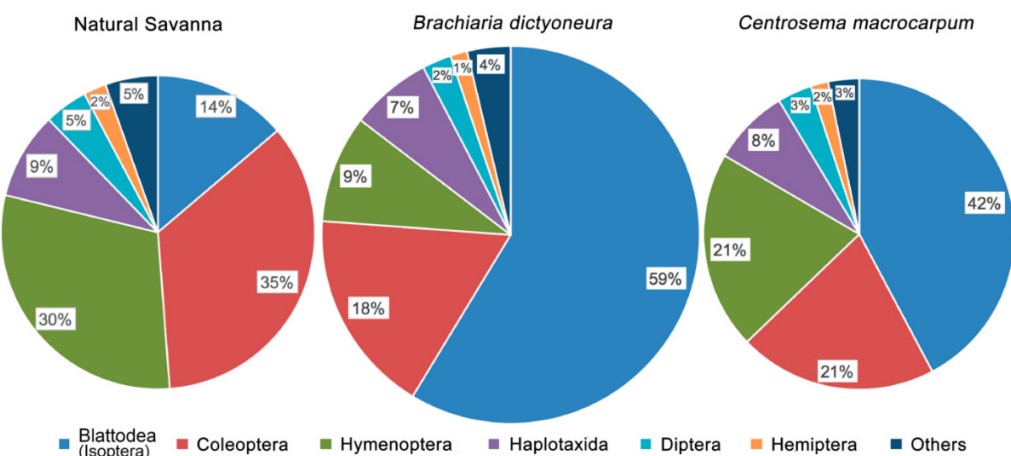

**Figure 3.** Relative density (%) of the orders of the soil macrofauna for every vegetation cover: natural savanna, maize crop associated with *Brachiaria dictyoneura* and *Centrosema macrocarpum* in the La Iguana station, Guárico state, Venezuela. Every color indicates a different taxon. Size of the circle represents the sum of the average density of every vegetation cover. Other: groups with average density less than 4 ind·m$^{-2}$.

### 3.2. Effect of Agroecological Management on the Vertical Distribution of Soil Macrofauna

The agricultural management of the savanna also affected the taxonomic structure of soil macrofauna in the vertical gradient. Figure 4 shows that the order of dominance of some taxa is altered with respect to NS and even the absence of some, such as Araneae in the soil layers of 0–5 and 5–15 cm of Bd, and the orders with a density of less than 1.2% in the layer of 5–15 cm of Cm. In this figure it can also be seen that the cultivation of maize associated with Bd had a greater impact, favoring the increase of six and eight times the abundance of Blattodea (Isoptera) in relation to the other orders. While excluding Isoptera, the culture associated with Cm showed a dominance distribution between orders more similar to NS.

The results on the taxonomic structure of soil macrofauna per plant cover suggest that the less favorable conditions for macrofauna were given by the cultivation of maize associated with Bd, while that associated with Cm presented similar traits to NS. The absence of some families of Coleoptera, Diptera and the order Lepidoptera in Bd and Cm could suppose that this group of insects had a high sensitivity to changes in the soil produced by the agroecological activity, while Termitidae was favored by this intervention.

These changes could be associated with variations in some edaphic properties (P and OM). For example, changes in P and OM were found, which presented lower average values ($p < 0.05$) in NS ($p = 2.5 \pm 1.7$ mg·kg$^{-1}$, OM = 0.96 ± 0.07) than in Bd ($p = 5.28 \pm 1.34$ mg·kg$^{-1}$, OM = 1.23 ± 0.18%) and even more in Cm ($p = 6.15 \pm 1.28$ mg·kg$^{-1}$, OM = 1.14 ± 0.18%). This was perhaps due to agricultural management [67,68]: on the one hand, since the P content increased due to the application of phosphate rock in the preparation of the land for crops; on the other hand, to the incorporation of OM to the soil, as a result of the vegetable residues that were left on the soil surface after the harvests. Several studies showed a positive relationship between OM and the soil macrofauna [69–71]. In this study, for example, it can be inferred that Termitidae (Blattodea: Isoptera) was widely favored over other invertebrates due to high OM content in cultivated soils. This corresponds to several authors [41,72] who attributed the dominance of Isoptera to the crop residues left on the soil.

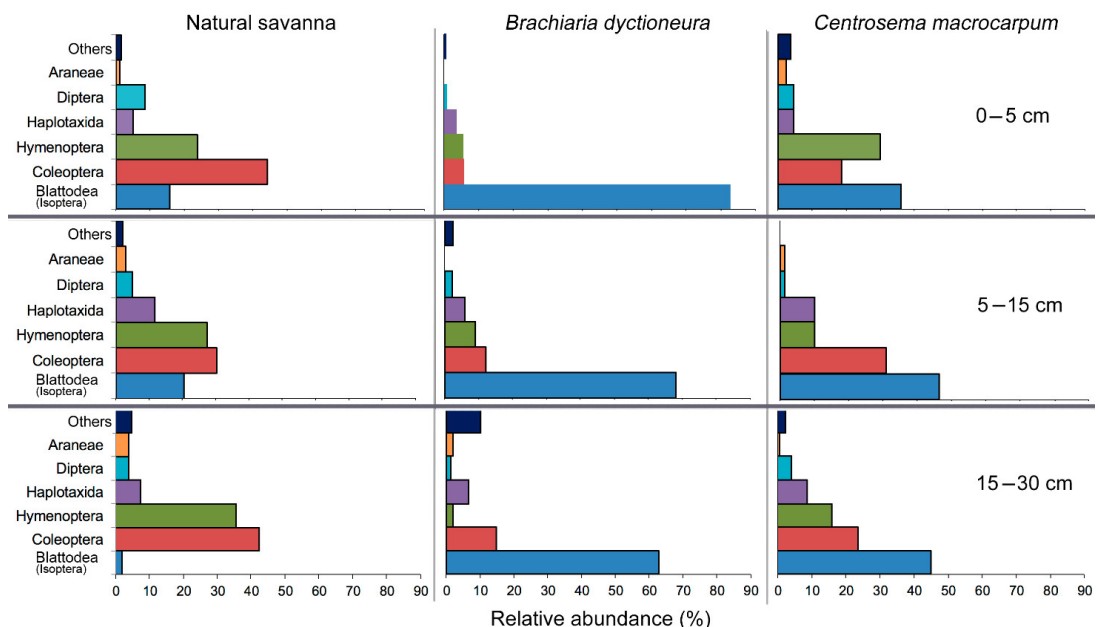

**Figure 4.** Relative abundance (%) of the soil macrofauna orders by soil depth (0–5, 5–15 and 15–30 cm) and by vegetation cover: natural savanna, maize crop associated with *Brachiaria dictyoneura* and *Centrosema macrocarpum*. Each color indicates a different taxon. Other: groups with relative average abundance less than 1.2%.

Another of the edaphic properties that showed variation between cover vegetal was BD, which presented the highest average value ($p$ <0.05) in NS ($1.6 \pm 0.07$ Mg·m$^{-3}$) with respect to Bd ($1.49 \pm 0.06$ Mg·m$^{-3}$) and Cm ($1.46 \pm 0.05$ Mg·m$^{-3}$). This difference suggests that agroecological management improved this condition, contradicting the reports of several authors [65,73–75] who found an increase of compaction of the soil in the plots of cultivation with respect to natural savannas.

However, agroecological management apparently did not have a significant effect on the ecological attributes of the soil macrofauna when their averages were compared between vegetation covers. This result could be due to the high variability of the data (coefficient of variation > 75%) when all sampling times were averaged. Even so, this contradicts previous works: on the one hand, those that report a positive effect of conservation management on soil macrofauna [10,40,76]; on the other hand, those who find a negative effect of conservation management applied in their studies (no tillage) [65,73,74]. On the contrary, when the sampling times were compared significant differences were found between them, which suggests that the climatic conditions exerted a greater effect than the changes produced by the agroecological activity.

### 3.3. Effect of Agricultural Management on Soil Macrofauna in the Temporal Gradient

As mentioned above, the average density of soil macrofauna was not affected by agricultural management. This was also corroborated by analyzing the effect of the disturbance on this density throughout the sampling times. The two-way analysis of variance, using the "vegetation cover" and "time" categories, indicated that there was no interaction between them ($F$: 1.18; $p = 0.37$) (Supplementary Material, Table S1), therefore, the temporal variation of the average soil macrofauna density occurred independently of the vegetation cover. Otherwise, in the analysis of the richness and diversity of families, a significant interaction was found between the categories "time" and "vegetation cover", so the temporal variation of these attributes was analyzed by discriminating by vegetation cover (Figure 5).

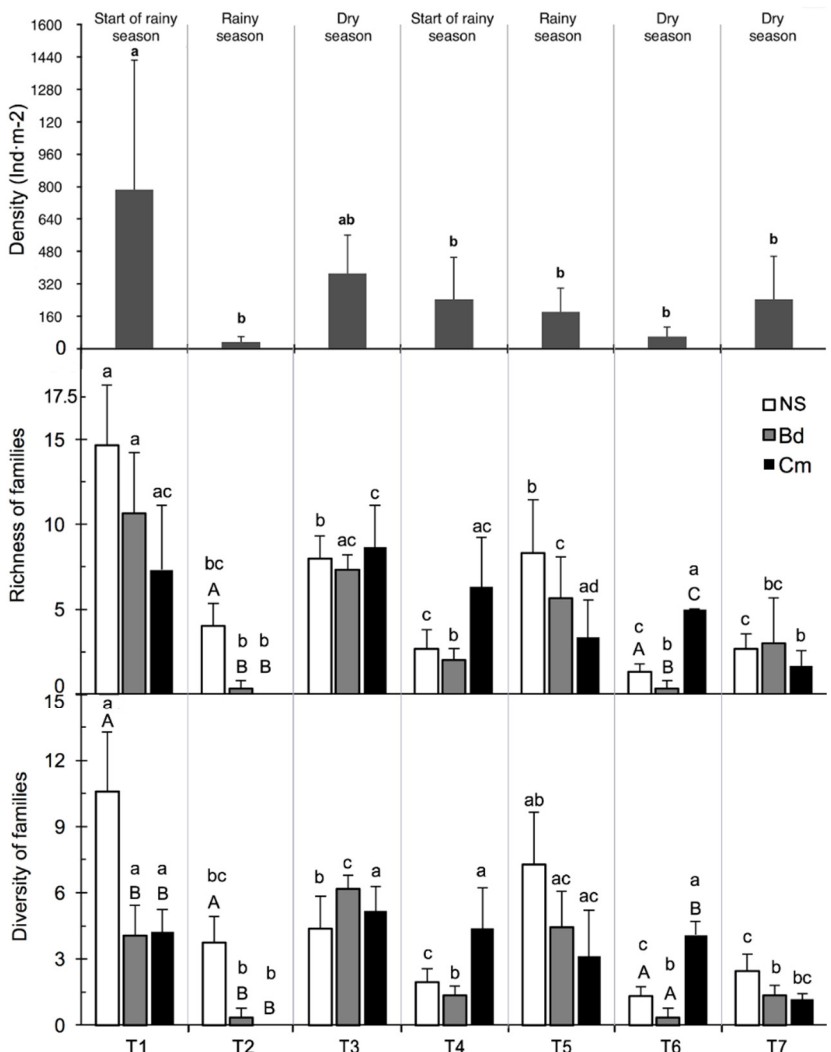

**Figure 5.** Average density of soil macrofauna without discrimination by coverings, regarding sampling times—T— (**upper part**); richness (**middle**) and diversity of families (**lower part**): natural savanna (NS), maize crop associate with *Brachiaria dictyoneura* (Bd) and *Centrosema macrocarpum* (Cm). Uppercase letters indicate differences ($p < 0.05$) between every vegetation cover for a same time (T), lowercase letters indicate differences between times for a same vegetation cover. Climatic seasons are indicated.

Figure 5 shows the temporal variation of the density (upper part), revealing that in T1 (beginning of the rainy season) the highest values were recorded ($p < 0.05$) ($785.33 \pm 643.04$ ind·m$^{-2}$), followed by T3 (dry season, with $371.55 \pm 193.16$ ind·m$^{-2}$). The lowest values were recorded in T2 (rainy season), with $32.0 \pm 26.2$ ind·m$^{-2}$. In addition, a decrease in density can be observed over time, corroborated through a linear correlation of Pearson ($r = -0.43$; $p < 0.05$).

Regarding the temporal variation of richness in each coverage, Figure 3 shows that NS had the highest values; T1 registered the maximum with $14.5 \pm 3.05$ families, while the lowest was T6 (dry season), with $1.33 \pm 1.12$ families. In the case of Bd, the temporal variation of richness registered the highest value in T1 ($10.52 \pm 3.87$ families) and the lowest in T2 and T6, both with $0.35 \pm 0.33$ families. In Cm, unlike NS and Bd, the highest value of richness was recorded in T3, with $8.67 \pm 3.05$ families, followed by T1 ($7.10 \pm 2.8$ families) and T5 (rainy season), with $3.33 \pm 2.8$ families. In T2, zero (0) families were recorded, being the sampling time and the vegetation cover where no individual of the soil macrofauna was found.

Concerning the variation of families' richness and diversity by coverage for each sampling time, Figure 3 shows that, for richness, there were only differences in T2 and T6 times, while diversity of families, in addition to these two times, was also in T1. In T2, NS presented the highest values ($p < 0.05$) of these attributes (richness: $4.0 \pm 1.3$ families and diversity: $3.7 \pm 1.2$ families), while in T6, Cm presented the highest values (richness: 5.0 families and diversity: $4.1 \pm 0.6$ families). In T1, the diversity of families also presented differences, with NS having the highest value reported ($10.6 \pm 2.7$ families).

The temporary variation of the soil macrofauna could be associated with the decrease ($p < 0.05$) in TP ($r = -0.55$), Ksat ($r = -0.44$) and the increase in Bd ($r = 0.56$), which was related ($p < 0.05$) to density of soil macrofauna (TP: $r = 0.46$; Bd: $r = -0.40$). This corresponds to the results found by other researchers [12,18,29,34] and is explained by the fact that with lower porosity apparent density increases ($r = -0.69$; $p < 0.05$) and hydraulic conductivity decreases ($r = 0.65$; $p < 0.05$). These changes in the physical structure of the soil negatively affected the exchange of gases and the infiltration of water with nutrients in the soil, which in turn affected the populations of edaphic invertebrates. It is important to highlight that, although the soil in this study is sandy, without natural compaction problems and with good infiltration capacity and water movement in the soil profile [46], our results suggest that these edaphic properties are sensitive to disturbance. This finding agrees with the observations of other authors [73], so it should be considered when planning agricultural activities on this type of soil.

When comparing the temporal variation of the ecological attributes of the soil macrofauna between the vegetal covers. Our results evidence an important reduction in the ecological attributes of soil macrofauna in the Bd and Cm of T2. This decrease could be associated with the negative impact of the establishment of the maize crop. This could explain the negative relation ($p < 0.05$) between this property and the ecological attributes of soil macrofauna (density: $r = -0.77$; family richness: $r = -0.75$; family diversity: $r = -0.76$) which contrasts with that reported in other investigation [77]. Likewise, a negative relationship was found between the families' richness and diversity of soil macrofauna with EC ($r = -0.87$ and $r = -0.88$, respectively), suggesting that the macrofauna of this soil is sensitive to its salinity. This result is contrary to that reported in previous studies [78], where a positive relationship was found between this property and the soil macrofauna.

*3.4. Relationship of Edaphic Macrofauna with Edaphic Properties*

The relationship between the edaphic properties and the density of the soil macrofauna families was evaluated through an RDA (Figure 6). The three main axes of this RDA explained 81% of the total variance. Axis 1 was determined mainly by Carabidae and Staphylinidae and in the opposite direction by Scarabaeidae and Formicidae. Axis 2 was determined by Glossoscolecidae and Formicidae and in the opposite direction by Termitidae. The third axis was determined by Termitidae and Glossoscolecidae and in the opposite direction by Tenthredinidae and Aphodiidae. According to the minimum correlation value, the soil properties that were significantly correlated with the RDA axes were TP, $P_{micro}$ and Ksat, EC and P (Supplementary Material, Table S2). Given the proximity between the vectors of these edaphic properties and the vectors of soil macrofauna, it can be infered that TP, $P_{micro}$ and Ksat were positively related to Glossoscolecidae, Staphylinidae and Carabidae and negatively related to Tenthredinidae, Tenebrionidae, Miridae and Scarabaeidae (first and second axis of the RDA). Meanwhile, Formicidae correlated positively with Pt and negatively with P, contrary to Termitidae, which correlated positively with P (second and third axes of the RDA).

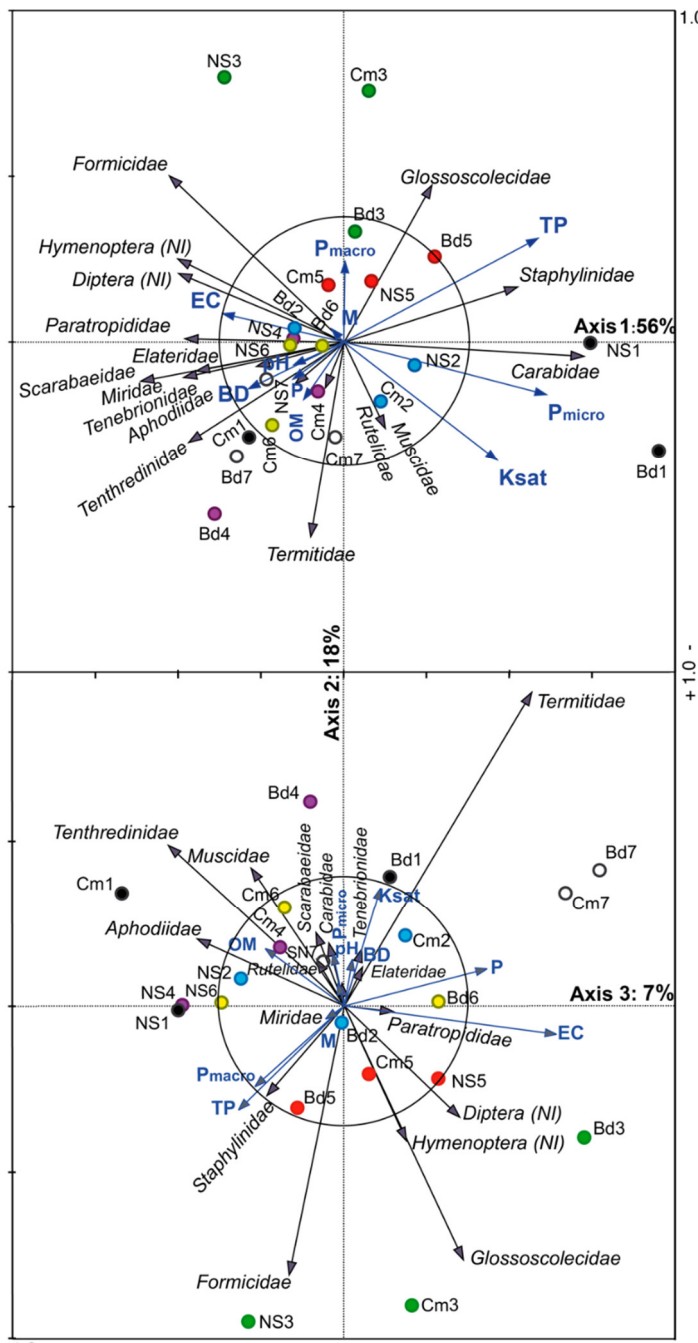

**Figure 6.** Ordering of the sampling sites (vegetation cover: natural savanna (NS)), maize crop associate with *Brachiaria dictyoneura* (Bd) and *Centrosema macrocarpum* (Cm) by sampling time (e.g., Bd1, NS3 . . . Bd$_X$, Cm$_X$ or NS$_X$; differentiated by color) according to the average density of the families of the soil macrofauna (italic writing) and their correlation with soil properties (blue vector) through an RDA. Axis 1 and 2: upper part; Axis 2 and Axis 3: lower part. Center circle: minimum correlation value (*r* = 0.38). Families with a density greater than 2 ind·m$^{-2}$ were considered. NI: Unidentified. EC: electric conductivity; P: available phosphorus; P$_{macro}$: macroporosity; P$_{micro}$: microporosity; TP: total porosity; BD: bulk density; M: percentage of humidity; Ksat: hydraulic conductivity.

In Figure 6 it can also be seen that, at least for these three axes, most of the sites tended to be grouped towards the center of the graph, which would indicate that there is a certain homogeneity in the density of the most important families of the Macrofauna. However, the RDA reveals that there was a certain ordering by sampling times, for example: in the

dimension formed by axes 2 and 3, sampling time 3 (T3: NS3, Bd3 and Cm3) differed from the rest, being more evident in the negative part of axis 2. These sites were associated with the dominance of Formicidae and Glossoscolecidae and low density of Termitidae. In the same direction, although more towards the center, the T5 sites (NS5, Bd5 and Cm5) were found. These sites from T3 and T5 were associated with elevated EC and TP values. In addition, the grouping of the sites sampled in T7 (Cm7 and Bd7) was found to be associated with a dominance of Termitidae and with high contents of P. In the dimension formed by axes 1 and 2, the NS1 and Bd1 sites were characterized by the predominance of Carabidae and Staphylinidae, associated with relatively high percentages of $P_{micro}$ and Ksat. In this same dimension, all the points corresponding to NS, ordered to the negative side of axis 3, were characterized by a relative dominance of Aphodiidae and low contents of P and EC.

Despite the non-interaction between the categories "time" and "vegetation cover", RDA evidenced a grouping of NS, which suggests a greater environmental homogeneity in these soils over time than in those intervened agroecologically. This phenomenon has also been recorded by other authors [10] who show greater heterogeneity in soils under cultivation as a result of crop rotation, types of cover and fertilizer input.

The correlation between Staphylinidae, Carabidae and Glossoscolecidae with Pmicro, Ksat and TP, and the latter with Formicidae, has also been reported by other authors [34,79,80]. We agree that the activity of these families increases with porosity and, therefore, with Ksat. Likewise, a positive correlation was found between P and CE with Diptera (NI), Hymenoptera (NI) and Termitidae larvae, as reported by several authors [77,78,81,82].

The differences between several of our findings and what has been reported in several works confirm what has been suggested by several authors [10,41,83]. They state that the different results that agricultural management can generate on the biotic and abiotic properties of the soil depend on a multiplicity of factors, such as characteristics of the management used, type of crop, soil, climate, vegetation and history of the use of the land among others. In other words, it is practically impossible to propose a standard management procedure for soil conservation in different ecosystems.

## 4. Conclusions

Under the conditions of this study and its duration of 2 years and 10 months, it can be concluded that the agricultural management used apparently did not affect the soil macrofauna. However, it favored Termitidae to the detriment of the equity of the community's taxonomic structure. It is important to highlight the need for longer studies (medium and long term) to evaluate the evolution of the effect of this type of conservation management on soil macrofauna.

Our results suggest that maize cultivation associated with the grass *Brachiaria dictyoneura* had a greater negative impact on soil macrofauna than maize cultivation associated with the legume *Centrasema macrocarpum*. Therefore, from the point of view of soil macrofauna, this vegetation cover was the most conservationist.

The fact of having found a temporal variation in the average of the density, richness of families and diversity of families of the soil macrofauna, independently of the vegetation cover, allows us to conclude that the effect of the climatic season was more important than agroecological management. This variation could be due to changes in the soil structure, so for this type of soil, despite being sandy loam, it should be an aspect to consider in the agricultural practices of this ecosystem.

Although there was no significant effect of the plant cover in several of the analyses carried out, a greater environmental homogeneity could be inferred in the soil of the NS. Likewise, the disappearance of Coccinelidae, Dynastidae (Coleoptera), Cecidyomidae (Diptera) and Pieridae (Lepidoptera) and the increase of Termitidae (Blattodea) with the agroecological management was reported, which indicated a high sensitivity of these taxonomic groups of soil macrofauna to this type of edaphic alteration.

**Supplementary Materials:** The following supporting information can be downloaded at: https://www.mdpi.com/article/10.3390/land11040464/s1, Table S1: summary of the effects of factorial ANOVA for the interaction between factors "sampling times" (Times) and "vegetal cover" (COV), using the average density of the soil macrofauna, its families richness and diversity; (https://doi.org/10.5281/zenodo.6299695, accessed on 25 February 2022) Table S2: correlation (*r*) of the physical-chemical properties of the soil with the axes of RDA, formed through the density of the families of the soil macrofauna of a savanna agroecosystem with maize crop associated with vegetal covers in the La Iguana station; (https://doi.org/10.5281/zenodo.6365727, accessed on 25 February 2022) Values in red indicate significant correlation ($p < 0.05$; $n = 21$).

**Author Contributions:** Conceptualization, J.A.M.-M. and R.M.H.-H.; methodology, J.A.M.-M., Z.L., R.M.H.-H. and R.J.-B.; formal analysis, J.A.M.-M. and G.K.S.; investigation, J.A.M.-M., R.M.H.-H., G.K.S. and I.C.; resources, J.A.M.-M., and G.K.S.; writing—original draft preparation, J.A.M.-M.; writing—review and editing, R.J.-B. and J.A.M.-M.; project administration, R.M.H.-H.; funding acquisition, R.M.H.-H. All authors have read and agreed to the published version of the manuscript.

**Funding:** This research was funded by Fondo Nacional para la Ciencia Innovación y Tecnología del Ministerio del Poder Popular para la Educación Universitaria, Ciencia y Tecnología of the Bolivarian Republic of Venezuela for the financial support of the project "Manejo Agroecológico de Suelos de Sabanas Bien Drenadas con Unidades de Producción Cereal-Ganado", grant number G-2002000398".

**Institutional Review Board Statement:** Not applicable.

**Informed Consent Statement:** Not applicable.

**Data Availability Statement:** Not applicable.

**Acknowledgments:** We want to express our gratitude to the "Fondo Nacional para la Ciencia Innovación y Tecnología del Ministerio del Poder Popular para la Educación Universitaria, Ciencia y Tecnología" of the Bolivarian Republic of Venezuela for the financial support of the project "Manejo Agroecológico de Suelos de Sabanas Bien Drenadas con Unidades de Producción Cereal-Ganado", N° G-2002000398, of which this study is part. Moreover, we thank the technical team of the Biogeochemistral Laboratory of IDECYT- UNERS for their support in the field and in the labs. Finally, we thank Antonio de Ascenção of the Animal Biology Laboratory, Science Faculty—ULA for his help in the identification of samples of macrofauna of the soil.

**Conflicts of Interest:** The authors declare no conflict of interest.

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
