# Peer review of "Effect of Maize Conservation Crops Associated with Two Vegetal Covers on the Edaphic Macrofauna in a Well-Drained Savanna of Venezuela"

_land, doi:10.3390/land11040464_

Round 1

Reviewer 1 Report

Paper is interesting, but there are some important mistakes in the classification of the fauna, and must be corrected. Authors must be review and add more relevante references to support their steatments. 

Author Response

Dear Editor and Reviewers,

We appreciate for your kind review of our manuscript. Thanks for your observations, corrections and suggestions.

Rev 1

Comments and Suggestions for Authors.

- Paper is interesting, but there are some important mistakes in the classification of the fauna, and must be corrected. Authors must be review and add more relevante references to support their steatments. 

Response: We are very grateful for your encouraging and the positive evaluation of our research. We have re-organized this manuscript to balance according to your comments. Thank you for your valuable suggestions.

-Line 46. Response: Thank you for the comments. Sanabria et al accepted. Done

-Line 53. Response: Thank you for the comments. Briones et al, is not accepted because this author makes a critical analysis and uses the same reference (Swift et al, 1979) to classify the soil fauna. Done.

-Line 64. Response: Thank you for the comments. Tauro et al accepted. Done

-Line 79. Response: Thank you for the comments. Morales et al. This is an interesting of work that I will review for reference where appropriate, in this manuscript or others in preparation. Thank you very much.

-Line 204. Response: Thank you for the comments. Done

-Table 3. The diversity index used was the Hill number (N1). This information can be found in section 2.7. for its calculation the number of families was used. For this reason, when "diversity" is indicated in the text, "diversity of families" is placed.

-Line 231. Response: Thank you for the comments. Have been considered and corrected accordingly

Again, we thank you very much for your comments and hope that this revised version will fit to your expectations. We wish you all the best!

Sincerely

Jimmy Morales

Reviewer 2 Report

The manuscript presents deals with conservationist agriculture, which is interesting. It is relevant and within the scope of the journal.

L 17 “The samples” what samples? soil? Plant?

The current abstract is not informative. This section should be rewritten carefully. Please add some numerical and statistical values to this section.

Table 1 how many soil samples?

What is CIC?

Soil pH=5 (strongly acidic), so many plant will not grew in this pH!

Some figures from the experimental should be added.

L 121 define “climatic times”

2.2. Design of the study: Till the end of this section, authors did not define the numbers of soil samples.

L 138: why authors used controlled burning method?

Result section should be reorganized, full description about the impact of different practice on soil properties should be added, which correspondence with “2.6. Laboratory analysis

L 298: “he RDA axes were TP, Pmicro….” From where authors get these results if they did not present it before?

Discussion:

L 333 It is better to drew a table to compare the output of this research with other research, instead of saying :….. as much lower than reported for humid tropical soil

L344 it is a result!’

In general, this section is not well organized, authors mixed between result and discussion. More interestingly, authors reported for many new results in this section!

This version is not informative! However, I see a merit in this research. Authors invited to reorganize their work in more appropriate and scientific way.  

Author Response

Dear Editor and Reviewers,

We appreciate for your kind review of our manuscript. Thanks for your observations, corrections and suggestions.

Rev 4

Comments and Suggestions for Authors.

The manuscript presents deals with conservationist agriculture, which is interesting. It is relevant and within the scope of the journal.

Response: We are very grateful for your encouraging and the positive evaluation of our research. We have re-organized this manuscript to balance according to your comments. Thank you for your valuable suggestions.

 L 17 “The samples” what samples? soil? Plant?

Response: Thank you for the comments: It is “soil and macrofauna”. We add it in this new manuscript version

The current abstract is not informative. This section should be rewritten carefully. Please add some numerical and statistical values to this section.

Response: Thank you for the comments : we have improved it

Table 1 how many soil samples?

Response: Thank you for the comments. 108 per depth. We put it in the table title in this new version.

What is CIC?

Response: Thank you for the comments. Sorry for the mistake, since it is expressed in Spanish: capacidad de cambio cationico. Obviously it is CEC, that is, Cation exchange capacity:  the total capacity of a soil to hold exchangeable cations. CEC is an inherent soil characteristic and is difficult to alter significantly. It influences the soil's ability to hold onto essential nutrients and provides a buffer against soil acidification.

Soil pH=5 (strongly acidic), so many plant will not grew in this pH!

Response: Thank you for the comments. The qualification that pH 5 is very strongly acidic depend of the author. Certainly for some authors it is not appropriate and we can change it to moderately acid or to acid. What if it is evident that there are numerous plants that grow at these pHs and others even much lower. In particular, one of the authors has analyzed the species Erica Ardevalensis at pHs close to 3 in Rio Tinto, Spain.

Some figures from the experimental should be added.

Response: Thank you for the comments. We have added some pictures.

L 121 define “climatic times”

Response: Thank you for the comments. Excuse us, we refer to “climatic seasons”. It was changed in the manuscript

2.2. Design of the study: Till the end of this section, authors did not define the numbers of soil samples.

Response: Thank you for the comments.  we add it. A mixed soil sample consisted of 4 soil sampling points per experimental unit. Therefore, per vegetation cover, there were 3 samples of mixed soil (12 sampling points) per each depth.

L 138: why authors used controlled burning method?

Response: Thank you for the comments. Reviewing, we realized that there is a writing error. In reality, there were no controlled burns in the natural savanna. What we did was respect the annual burning that occurs naturally in these savannas and that is what defines that ecosystem Gasson et.al 2014. The control was to avoid burning in cultivated plots. We control it with fire barriers. We corrected that error in the new manuscript.

Gasson, R.; Butt-Colson, A.; Leal, A.; Bilbao, B. Ecología Histórica de La Gran Sabana (Estado Bolivar, Venezuela) Entre Los Siglos XVIII y XX. In Guyanas e Orinoco; Rostain, S., Ed.; Instituto Francés de Estudios Andinos: Lima, Perú, 2014; pp 113–121.

Result section should be reorganized, full description about the impact of different practice on soil properties should be added, which correspondence with “2.6. Laboratory analysis”

Response: Thank you for the comments. But, If we accept the reviewer's suggestion we oppose the positive review of the other 3 reviewers, so we don't know what to do at this point.

L 298: “he RDA axes were TP, Pmicro….” From where authors get these results if they did not present it before?

Response: Thank you for the comments. This work is part of a group project. The soil results corresponded to other colleagues and in this publication we can only place the analyzes with their raw data. However, he left in the supplementary material a table with the correlation analysis with the axes of the RDA

Link : https://doi.org/10.5281/zenodo.6365727

Discussion:

L 333 It is better to drew a table to compare the output of this research with other research, instead of saying :….. as much lower than reported for humid tropical soil

Response: Thank you for the comments. We appreciate the reviewer's proposal, but this issue will be addressed in a new and broader work that is practically finished, which we will submit for publication.

L344 it is a result!’

Response: Thank you for the comments. changed

In general, this section is not well organized, authors mixed between result and discussion. More interestingly, authors reported for many new results in this section!

Response: Thank you for the comments.  Following your suggestion we accept it, for which we join results and discussion.

This version is not informative! However, I see a merit in this research. Authors invited to reorganize their work in more appropriate and scientific way.

Thank you for your valuable suggestions.

Reviewer 3 Report

My comments on the manuscript “Effect of maize conservation crops, associated with two vegetal covers on the edaphic macrofauna in a well-drained savannah of Venezuela”, which has been submitted to Land journal, are presented below.

The manuscript is very interesting. The Authors comprehensively presented the problem the response of the soil macrofauna to the establishment of maize conservation crops (Zea mays), associated with Brachiaria dictyoneura and Centrosema macrocarpum.

Please complete the information in the methodology and specify the row spacing for maize. Please also state in which years the research was carried out.

I recommend this paper [Manuscript ID: land-1635884] for publication in Land journal.

Author Response

Dear Editor and Reviewers,

We appreciate for your kind review of our manuscript. Thanks for your observations, corrections and suggestions.

REV 2

Comments and Suggestions for Authors

My comments on the manuscript “Effect of maize conservation crops, associated with two vegetal covers on the edaphic macrofauna in a well-drained savannah of Venezuela”, which has been submitted to Land journal, are presented below.

 The manuscript is very interesting. The Authors comprehensively presented the problem the response of the soil macrofauna to the establishment of maize conservation crops (Zea mays), associated with Brachiaria dictyoneura and Centrosema macrocarpum.

Please complete the information in the methodology and specify the row spacing for maize. Please also state in which years the research was carried out.

 I recommend this paper [Manuscript ID: land-1635884] for publication in Land journal.

Response: We are very grateful for your encouraging and the positive evaluation of our research. We have re-organized this manuscript to balance according to your comments. Thank you for your valuable suggestions.

Regarding the question on the information in the methodology it has been done. The the row spacing for maize is 18 m. And finally the investigation was carried out from 2005 to 2008. This information has been adde in the text (section 2.2. Design of the study).

Again, we thank you very much for your comments and hope that this revised version will fit to your expectations. We wish you all the best!

Sincerely

Jimmy Morales

Reviewer 4 Report

Dear Authors,

I think that the manuscript is valuable and contains important information on soil biota related to agricultural management.

I marked some minor issues.

I suggest an English check!

Lines 413-414: The conclusion that „The results of this study partially refute the hypothesis that agroecological management would favor the community of soil macrofauna . . . ” needs to be further elaborated due to the fact that the results are not originated long-term analyses, so authors cannot conclude that agroecological management does not favor the community of soil macrofauna, I mean, not „in general”. You could only conclude this after several years of investigation I think. I would just suggest adding that it was concluded only after a 3-year experiment. We do not know what would happen after 10 or 20 or 30 years of different management. Especially if extensive use of chemicals were applied on all the areas used for the experiments.

Otherwise, I congratulate on your work!

Best regards, Reviewer X

Author Response

Dear Editor and Reviewers,

We appreciate for your kind review of our manuscript. Thanks for your observations, corrections, and suggestions. 

REv 3

Comments and Suggestions for Authors

Dear Authors,

I think that the manuscript is valuable and contains important information on soil biota related to agricultural management.

I marked some minor issues.

I suggest an English check!

Lines 413-414: The conclusion that „The results of this study partially refute the hypothesis that agroecological management would favor the community of soil macrofauna . . . ” needs to be further elaborated due to the fact that the results are not originated long-term analyses, so authors cannot conclude that agroecological management does not favor the community of soil macrofauna, I mean, not „in general”. You could only conclude this after several years of investigation I think. I would just suggest adding that it was concluded only after a 3-year experiment. We do not know what would happen after 10 or 20 or 30 years of different management. Especially if extensive use of chemicals were applied on all the areas used for the experiments.

Otherwise, I congratulate on your work!

Response: We are very grateful for your encouraging and the positive evaluation of our research. We have re-organized this manuscript to balance according to your comments. Thank you for your valuable suggestions. Conclusions has been changed according to your comments.

About: what was the situation in the range of 5 to 50 micrometer? Response: we do not consider it.

Again, we thank you very much for your comments and hope that this revised version will fit to your expectations. We wish you all the best!

Sincerely

Jimmy Morales

Round 2

Reviewer 2 Report

The current version is better than the previous one. I think the result of soil samples are essential for this research. However, authors reported that they don’t have these data.

Fig. 1, 2 should be added as supplementary. It will be good if author add a flowchart to describe their work.